# Treatment with Bixin-Loaded Polymeric Nanoparticles Prevents Cigarette Smoke-Induced Acute Lung Inflammation and Oxidative Stress in Mice

**DOI:** 10.3390/antiox11071293

**Published:** 2022-06-29

**Authors:** Alexsandro Tavares Figueiredo-Junior, Samuel Santos Valença, Priscilla Vanessa Finotelli, Francisca de Fátima dos Anjos, Lycia de Brito-Gitirana, Christina Maeda Takiya, Manuella Lanzetti

**Affiliations:** 1Instituto de Ciências Biomédicas, Universidade Federal do Rio de Janeiro, Rio de Janeiro 21941-902, Brazil; figueiredojunior.at@ufrj.br (A.T.F.-J.); samuelv@icb.ufrj.br (S.S.V.); fatimadosanjoss@ufrj.br (F.d.F.d.A.); lyciabg@histo.ufrj.br (L.d.B.-G.); 2Departamento de Produtos Naturais e Alimentos da Faculdade de Farmácia, Universidade Federal do Rio de Janeiro, Rio de Janeiro 21941-902, Brazil; finotelli@pharma.ufrj.br; 3Instituto de Biofísica Carlos Chagas Filho, Universidade Federal do Rio de Janeiro, Rio de Janeiro 21941-902, Brazil; takiyacm@biof.ufrj.br

**Keywords:** bixin, polymeric nanoparticle, lung inflammation, antioxidant, cigarette smoke, Nrf2

## Abstract

The use of annatto pigments has been evaluated as a therapeutic strategy in animal models of several health disorders. Beneficial effects were generally attributed to the inhibition of oxidative stress. Bixin is the main pigment present in annatto seeds and has emerged as an important scavenger of reactive oxygen (ROS) and nitrogen species (RNS). However, this carotenoid is highly hydrophobic, affecting its therapeutic applicability. Therefore, bixin represents an attractive target for nanotechnology to improve its pharmacokinetic parameters. In this study, we prepared bixin nanoparticles (npBX) and evaluated if they could prevent pulmonary inflammation and oxidative stress induced by cigarette smoke (CS). C57BL/6 mice were exposed to CS and treated daily (by gavage) with different concentrations of npBX (6, 12 and 18%) or blank nanoparticles (npBL, 18%). The negative control group was sham smoked and received 18% npBL. On day 6, the animals were euthanized, and bronchoalveolar lavage fluid (BALF), as well as lungs, were collected for analysis. CS exposure led to an increase in ROS and nitrite production, which was absent in animals treated with npBX. In addition, npBX treatment significantly reduced leukocyte numbers and TNF-α levels in the BALF of CS-exposed mice, and it strongly inhibited CS-induced increases in MDA and PNK in lung homogenates. Interestingly, npBX protective effects against oxidative stress seemed not to act via Nrf2 activation in the CS + npBX 18% group. In conclusion, npBX prevented oxidative stress and acute lung inflammation in a murine model of CS-induced acute lung inflammation.

## 1. Introduction

Chronic obstructive pulmonary disease (COPD) is the third main cause of death worldwide [1,2,3,4,5]. Its pathophysiology is characterized by chronic inflammation as well as the progressive and irreversible destruction of the lung parenchyma, with enlargement of alveolar air spaces due to alveolar leukocyte activation, decreased α1-antitrypsin expression and secretion of metalloproteinases (e.g., MMP-9) [6,7], collectively causing obstructed airflow from the lungs [8]. The major risk factor for COPD is chronic cigarette smoke (CS) exposure [9]. Importantly, short-time CS exposure is already capable of promoting oxidative stress and acute lung inflammation (ALI) [10,11,12,13,14,15,16]. Although ALI does not include all pathogenic aspects of COPD, the two conditions certainly share common features, including an increase in reactive oxygen species (ROS) production due to an imbalance between oxidants and antioxidants in response to CS exposure and associated inflammatory events.

CS contains many free radicals and oxidant molecules (alkyl, peroxyl, nitric oxide and superoxide anion) as well as semiquinone-derivative compounds in the particulate phase [17]. Exposition to these molecules can lead to the activation of alveolar macrophages and recruitment of neutrophils, which in turn stimulates resident and infiltrated leukocytes to produce ROS and inflammatory mediators. This supports the notion that CS exposure results in lung injury, at least in part through the induction of a redox imbalance [10]. Oxidative stress induces cellular damage, affecting proteins via tyrosine nitration [18], induces cell apoptosis through lipid peroxidation [19] as well as caspase activation [20], and increases leukocyte recruitment to the pulmonary parenchyma by upregulating cytokines such as tumor necrosis factor alpha (TNF-α) [21].

Nuclear factor-erythroid 2 related factor 2 (Nrf2) is one the most functionally important transcription factors related to the endogenous antioxidant defense system. It regulates the transcription of various enzymes, including superoxide dismutase (SOD) and catalase (CAT), that play a crucial role in the neutralization of ROS [22]. Previous studies have shown that acute CS exposure can cause a redox imbalance through Nrf2 pathway dysregulation, which could promote oxidative stress [21].

*Bixa orellana* L., belonging to the Bixaceae family and popularly known as annatto, is a native plant from tropical America that contains various carotenoid derivatives, terpenoids, tocotrienols and flavonoids [23]. *B. orellana* L. extracts are widely used as a natural food and textile colorant and are also increasingly considered by the cosmetic industry as a red color additive and/or antioxidant [24]. The use of annatto pigments has been evaluated as a therapeutic strategy in animal models for several health disorders [25,26,27,28,29,30,31,32,33,34,35]. In general, the beneficial effects of these treatments were associated with antioxidant properties.

We previously demonstrated the functional involvement of oxidative substances in the development of CS-induced ALI. For the current work, we decided to use the same CS-induced murine model of ALI to study the effects of bixin. Bixin is the main pigment present in annatto seeds, comprising up to 80% of total carotenoids, and has emerged as an important scavenger of reactive oxygen (ROS) and nitrogen species (RNS) [33,36]. However, the long carbonic chain makes bixin highly hydrophobic and affects its bioavailability. Therefore, bixin is an attractive target for nanotechnology in order to improve its pharmacokinetic parameters and fully explore the potential therapeutic benefits.

In the current study, we prepared bixin-loaded polymeric nanoparticles and evaluated their effects on indices of inflammation and oxidative stress using a murine model of CS-induced ALI.

## 2. Materials and Methods

### 2.1. Materials

Annatto seeds were purchased from Florien Fitoativos (São Paulo, SP, Brazil). Sodium hydroxide (NaOH), trichloroacetic acid (TCA), thiobarbituric acid (TBA), 1,1,3,3-tetramethoxypropane (TMP), hydrochloric acid (HCl), sodium chloride (NaCl), potassium hydroxide (KOH), sodium dihydrogen phosphate (NaH_2_PO_4_), acetic acid glacial (Hac), sodium hydrogen phosphate (Na_2_HPO_4_), gelatin from bovine skin, Coomassie Brilliant blue G 250, zinc chloride (ZnCl_2_), calcium chloride (CaCl_2_), bovine serum albumin (BSA), potassium phosphate dibasic (K_2_HPO_4_), potassium phosphate monobasic (KH_2_PO_4_), sodium acetate, polycaprolactone (average MW ~14,000, average Mn ~10,000 by GPC), poloxamer 188 (average Mn ~8400), paraffin, Tris-Base, Triton X-100, β-actin (AC-15, mouse anti-mouse, A5441) primary antibody, glycerol, sodium dodecyl sulfate (SDS), hexadecyltrimethylammonium bromide (HTAB), ECL™, tetramethylbenzidine (TMB) and hydrogen peroxide were purchased from Sigma-Aldrich (St. Louis, MO, USA). Chloroform, acetonitrile (ACN), methanol (MeOH), dichloromethane (DCM), dimethyl sulfoxide (DMSO), acetone and formaldehyde were purchased from Merck (Darmstadt, Germany). Tween-80 and n-hexane were purchased from Vetec (Rio de Janeiro, RJ, Brazil). Coloração panótico rápido (620529) was purchased from LB Laborclin (Curitiba, PR, Brazil). Nuclear factor erythroid 2-related factor 2 (Nrf2) (C-20, rabbit anti-mouse, sc-722), Kelch ECH associating protein 1 (Keap1) (H-190, rabbit anti-mouse, sc-33569) and nitrotyrosine (PNK) (B-5, rabbit anti-mouse, sc-55256) primary antibodies and donkey anti-rabbit IgG-HRP (sc-2020) secondary antibody were purchased from Santa Cruz Biotechnology Inc (Dallas, TX, USA). Goat anti-mouse IgG-HRP (31430) secondary antibody and tumor necrosis factor alpha (TNF-α) mouse ELISA kit (BMS607-3) were purchased from Thermofisher Scientific (Waltham, MA, USA). Acrylamide/bis solution (30%, 29:1 (1610156)), polyvinylidene difluoride (PVDF) membranes and quick start™ Bradford 1x dye reagent (5000205) were purchased from Bio-Rad Laboratories (Tokyo, Japan).

### 2.2. Nanoparticle Preparation

Bixin was isolated from annatto seed and incorporated onto polymeric nanoparticles (npBX) following a methodology previously developed by our research group and described by Figueiredo-Junior, 2020 [37]. This procedure is based on an interfacial deposition method [38,39]. Briefly, an aqueous phase with 10% (*w*/*v*) poloxamer 188 and an organic phase consisting of 10% polycaprolactone (*w*/*v*) and 1% bixin (*w*/*v*) in acetone were prepared and added to the aqueous phase under magnetic stirring at 40 °C (5-3—Fisatom, São Paulo, SP, Brazil) until acetone was completely evaporated. To generate blank nanoparticle (npBL), the same methodology was followed without bixin. The nanoparticle (NP) suspension was filtered (0.44 µM Millipore^®^ filter) to remove any non-encapsulated bixin or polycaprolactone residues and then freeze-dried (Freeze Dry Systems, Labconco). The dried NPs were resuspended to 6, 12 and 18% (*w*/*v*) in PBS buffer.

### 2.3. Bixin Content

The dry annatto seeds extract was analyzed by ultra-high-performance liquid chromatography (UHPLC) (THERMOFISHER SCIENTIFIC, Dionex Ultimate 3000) at 457 nm using a ReproSil Gold C18 column (5 µm, 4.6 × 250 mm, Dr Maisch GmbH, Ammerbuch, Germany) as the stationary phase and ACN: MeOH: DCM (43:43:14) as the mobile phase in an isocratic mode at 30 °C with a flow rate of 1 mL/min [37,40] to determine the extraction yield. The result was expressed as the percentage of chromatogram integrated peaks reflecting the average of triplicate measurements. Chromatograms were obtained with Chromeleon™ 6.8 Chromatography Data System (CDS) Software (v.6.8, Thermofisher Scientific, Waltham, MA, USA).

### 2.4. Physical Characterization of Nanoparticles

The mean hydrodynamic diameter (MHD) and polydispersity index (PDI) of NPs were determined by dynamic light scattering (DLS, 90Plus/BI_MAS, Brookhaven Instruments Corp., Holtsville, NY, USA) at 25 °C with a scattering angle of 90° in a neutral pH aqueous solution. Zeta potential (ZP) was measured by a Zeta Plus Analyzer (Zeta Plus, Brookhaven Instruments, Holtsville, NY, USA) using the electrophoretic mode under neutral pH. To study NP morphology, particles were subjected to transmission electron microscopy (TEM) and analyzed by JEOL 1200 EX at 80 kV in a 300-mesh copper grid (Ted Pella, Inc, Redding, CA, USA) without staining. Mean diameter (MD) was calculated by considering all NPs as spheres and measuring size (while taking the scale bar into consideration) in 10 different, random fields.

### 2.5. Nanoparticle Thermal Stability

Thermogravimetric analysis (TGA) and differential scanning calorimetry (DSC) were performed on npBX and npBL. Samples were placed in an alumina crucible and subjected to thermal gravimetric analysis (Shimadzu^®^, model TGA 51, Kyoto, Japan) with a temperature range from 27 °C up to 1200 °C with an incremental increase in heat of 10 C/min in an inert nitrogen atmosphere at a gas flow of 50 mL/min. For DSC analysis (Shimadzu^®^, model DSC 60 PLUS), samples were packed into an aluminum pan, heated up to 90 °C, cooled to room temperature and then heated again up to 600 °C (with the heat increasing at a rate of 10 °C per minute) in an inert nitrogen atmosphere at 50 mL/min.

### 2.6. Animals

Eight-week-old *Mus musculus* (C57BL/6) male mice (20–25 g) were purchased from the Multidisciplinary Center for Biological Investigation on Laboratory Animal Science (CEMIB—UNICAMP, Campinas, Brazil) and fed ad libitum with Purina chow (Nuvilab^®^, Curitiba, Brazil) with unrestricted access to water in a controlled environment maintained at 22 ± 2 °C, 50–70% relative humidity and a 12 h light/dark cycle (EB-273B, Insight Equipamentos Ltda, Ribeirão Preto, SP, Brazil). Mice were acclimatized for two weeks prior to experimental procedures. The local Animal Ethics Committee approved all experimental procedures (CEUA/CCS/UFRJ-096/19).

### 2.7. Experimental Procedure

Mice were divided into 5 experimental groups: exposed to ambient air (AA, sham smoked) and treated with 18% (*w*/*v*) npBL suspension (AA + npBL); exposed to CS and treated with 18% (*w*/*v*) npBL suspension (CS + npBL); and exposed to CS and treated with npBX in three different concentrations (CS + npBX 6%, CS + npBX 12% and CS + npBX 18% (*w*/*v*)). The treatment was administered daily by gavage in a total volume of 100 µL each in order to evaluate if the colloidal suspension of npBX would retain bixin antioxidant properties. Animals were exposed to CS on 5 consecutive days according to a conventional protocol [10,16,37]. Briefly, mice were placed in an inhalation chamber (40 cm long, 30 cm wide, 25 cm high) inside an exhaustion chapel. Puffs from cigarettes were subsequently expelled into the exposure chamber, and animals were exposed for 6 min. The cover chamber was then lifted, and the smoke was completely removed within 1 min by turning on the exhaust fan of the chapel; this procedure was repeated for each cigarette. Mice were exposed to a total of 12 cigarettes per day, 4 in the morning, 4 in the midday, and 4 in the afternoon. CS exposure was performed using full-flavored Marlboro cigarettes (10 mg tar, 0.9 mg nicotine, and 10 mg carbon monoxide). Every day between the CS exposure, before the second inhalation round, animals received the suspension of npBL or npBX by oral gavage in a total volume of 100 µL.

### 2.8. Bronchoalveolar Lavage

On the sixth day, mice were sedated by isoflurane inhalation and immediately euthanized by cervical dislocation. Bronchoalveolar lavage fluid (BALF) was obtained by inserting a cannula in the trachea and washing the lungs three times with 500 μL 0.9% (*w*/*v*) NaCl saline solution at 4 °C, rendering a total of 1.5 mL per animal, which was conserved on ice. BALF was centrifuged at 600× *g* for 10 min (Z326 Hermle, Sayreville, NJ, USA); the supernatant was kept at 4 °C for posterior analysis and the pellet containing the inflammatory cells (alveolar leukocytes) was resuspended in ice-cold saline solution. A Neubauer chamber was used to count total cell numbers. Differential cell counts were performed in cytospin preparations stained with Panotico, counting 100 cells per lamina considering morphological criteria (neutrophils, macrophages or lymphocyte cells).

### 2.9. Tissue Processing and Lung Homogenate Preparation

After BALF recovered, the left lungs of all mice were sagittally divided in half. The proximal part containing the pulmonary hilum was fixed for 48 h with 10% (*v*/*v*) phosphate-buffered formalin (pH 7.2) and then embedded in paraffin. Sagittal 4 µm serial sections of the left lung were stained with hematoxylin and eosin (H&E) for histological analysis and prepared for immunohistochemistry. The distal part was homogenized in 500 µL zymography lysis buffer (0.5 M tris-base and 0.2% (*v*/*v*) Triton X-100 pH 7.4).

Right lungs were also divided into two parts (part 1: post-caval and upper; part 2: median and lower). The upper part was homogenized in 700 µL of lysis buffer (1 tablet of protease inhibitor, 0.1% Triton X-100 in 100 mL of phosphate buffer pH 7.4) for immunological analyses, and the median/lower lobes in 500 µL potassium phosphate buffer 10 mM pH 7.4 for biochemical analysis.

### 2.10. Myeloperoxidase (MPO) Activity

MPO activity in BALF was measured with the HTAB, TMB and hydrogen peroxide method. Samples (100 µL) were centrifuged with 900 µL HTAB at 14,000× *g* for 15 min. The supernatant (75 µL) was collected and incubated with 5 µL TMB for 5 min at 37 °C. Hydrogen peroxide (50 µL) was added, and the mixture was incubated for 10 min at 37 °C. Then, 125 µL sodium acetate buffer was added, and the OD of the solution was measured at 630 nm with a microplate reader UV/Vis spectrophotometer (Molecular Devices Spectra Max 250 microplate reader, San Jose, CA, USA) [41].

### 2.11. Malondialdehyde (MDA) Products and Nitrite Quantification

MDA levels were determined according to the thiobarbituric acid reactive substances (TBARS) method as described by Draper and Hadley (1990) [42]. First, proteins from lung homogenates were precipitated with 10% (*w*/*v*) TCA and centrifuged at 900 g for 10 min. The supernatant was mixed with 0.67% (*w*/*v*) TBA and heated for 15 min at 100 °C. The resultant solution was extracted with n-butanol, and the organic phase was measured in a UV/Vis spectrophotometer microplate reader (Molecular Devices Spectra Max 250 microplate reader, San Jose, CA, USA) at 532 nm. A calibration curve was plotted as TMP concentration against OD (R^2^ = 0.9976).

Nitrite detection followed the Griess reaction methodology. A 2.3 mmol sulfanilamide in 3% (*v*/*v*) phosphoric acid aqueous solution and a 0.2 mmol N-1-napthylethylenediamine dihydrochloride in 3% (*v*/*v*) phosphoric acid aqueous solution were prepared (solutions 1 and 2, respectively). Approximately 100 µL of BALF samples were mixed with 50 µL of each of these solutions in a microplate and incubated at room temperature for 30 min. Next, the OD of the solution was measured at 540 nm with a microplate reader UV/Vis spectrophotometer (Molecular Devices Spectra Max 250 microplate reader, San Jose, CA). A calibration curve was plotted as different sodium nitrite concentrations against OD (R^2^ = 0.9988).

### 2.12. Superoxide Dismutase (SOD) and Catalase (CAT) Activity

SOD activity was assayed by monitoring the inhibition of epinephrine auto-oxidation at 480 nm [43], and CAT activity was measured by the rate of decrease of hydrogen peroxide at 240 nm [44] in lung homogenates.

### 2.13. Western Blotting

The total protein content of each sample was quantified following the Bradford method. Equal amounts of protein (30 µg) were resolved in sodium dodecyl sulfate–polyacrylamide gel electrophoresis (SDS-PAGE) and electrotransferred to PVDF membranes, which were subsequently blocked with 3% (*w*/*v*) BSA solution. Next, the membranes were incubated with primary antibodies against PNK (overnight at 4 °C) and β-actin (for 2 h at room temperature). After washing and exposure to the appropriate horseradish peroxidase-conjugated secondary antibody, protein bands were visualized using an ECL detection system and quantified (relative to β-actin) by ImageJ software (GitHub, San Francisco, CA, USA).

### 2.14. Enzyme-Linked Immunosorbent Assay (ELISA) to Detect TNF-α Levels

TNF-α was quantified in BALF samples using a specific ELISA kit containing a rat anti-mouse monoclonal antibody and with a detection limit of 2000 pg/mL, according to the manufacturer’s instruction. A calibration curve was plotted using different mouse recombinant TNF-α standard concentrations against OD (R^2^ = 0.9999).

### 2.15. Gelatin Zymography

Protein (30 µg) was resolved in 8% SDS-PAGE containing 1.5 mg/mL gelatin for 3 h (100 V at 4 °C). The gel was subjected to a protein renaturation process by incubating first in 2.5% (*v*/*v*) Triton X-100 solution for 45 min at room temperature and then in a developing buffer (50 mM Tris-HCl pH 8.0, 10 mM CaCl_2_, 2 µM ZnCl_2_) for 24 h at 37 °C. After, gels were stained with 0.05% (m/v) Coomassie blue solution for 1 h and then appropriately destained with MeOH:HAc (50:50). MMP-9 activity was quantified using ImageJ software (MD, USA).

### 2.16. Immunohistochemistry

Lung sections were blocked with 3% BSA solution, washed in PBS-Triton X-100 2.5% (*v*/*v*), and subsequently incubated with 10% (*v*/*v*) hydroperoxide in PBS. Next, sections were incubated with a primary antibody against Nrf2. After the incubation and washing steps, the attached primary antibody was linked to dextran polymer according to the manufacturer’s protocol (Envision kit, Dako, Carpinteria, CA, USA) and the final reaction was achieved by immersing the sections in a solution of 3,30-diaminobenzidine (DAB). Finally, the sections were counterstained with hematoxylin and images were captured at 40× magnifications. The percentage of Nrf2-positive areas was obtained by automatized counting using ImageJ software for all positive areas in 10 fields per group for the staining.

### 2.17. Statistical Analysis

Data are expressed as mean ± SEM. Statistical significance of differences was evaluated by one-way ANOVA followed by Bonferroni post-test and then to outliers test. GraphPad Prism software was used to perform statistical analyses (version 8.0, CA, USA). Differences were considered to be statistically significant when *p* < 0.05.

## 3. Results

### 3.1. Bixin Extraction

Extraction rendered a single purified substance with absorption at 457 nm, as shown by the one peak with a retention time of 3.040 min in the chromatogram (Figure 1). Based on our previous experience with this extract, we could confirm that this represented bixin [37]. The process yield was 98%.

### 3.2. Nanoparticle Characterization

#### 3.2.1. Size, Zeta Potential and External Morphology

Table 1 include data obtained from DLS and TEM analyses. Mean hydrodynamic diameter (MHD), polydispersity index (PDI) and zeta potential (ZP) showed acceptable values, comparable to those described in our previous work, confirming that the NP preparation process is reproducible. The npBX had an MHD of 23.57 ± 0.17 nm with PDI 0.23 ± 0.01, indicating a homogeneous population with low diameter variability and a ZP of −31.73 ± 10.10 mV.

Figure 2A visually confirms a homogeneous nanoparticle population with low diameter variability, which corroborates the PDI data obtained from DLS. TEM analysis (Figure 2B–D) demonstrated a spherical morphology of these nanoparticles, with a diameter of 6.51 ± 0.34 nm.

#### 3.2.2. Thermal Stability of Nanoparticles

The npBL TGA/DrTGA analysis (Figure 3A) showed a mass reduction in four stages in a temperature range of 27 °C to 800 °C. The first DrTGA peak occurred at 249.05 °C, indicating 43.87% loss, the second peak at 350.15 °C showed 29.09% loss, the third peak at 455.57 °C demonstrated 19.75% loss and the last DrTGA peak at 499.71 °C represented 4.74% loss of mass. The TGA/DrTGA for npBX (Figure 3B) showed two stages of loss of mass, the first occurring with a triple peak indicating 95.39% loss and the second stage consisting of a DrTGA peak with a loss of 1.68%. The second stage had 1.68% weight loss with a DrTGA peak at 511.67 °C. These results suggest that bixin stabilizes the polymeric NP matrix, considering it reduces the number of degradation stages and the extent of nanoparticle degradation in response to heat.

DSC analysis (Figure 3C) showed no incompatibility between bixin and the polymeric matrix. PCL and P188 both showed an endothermic peak near 60 °C, characteristic of their melting points, which was preserved when bixin was incorporated in the formulations, suggesting that this product has a first degradation around 60 °C. The npBL presented an exothermic peak at 250 °C, which disappeared with bixin.

### 3.3. npBX Prevents ALI Induced by CS Exposure

#### 3.3.1. npBX Preserves Pulmonary Histoarchitecture

As anticipated, the lung parenchyma was preserved in animals exposed to ambient air and npBL (Figure 4A), with only a few resident leukocytes, indicating npBL administration did not disrupt basal morphology or cause notable inflammation. Figure 4B shows the lung parenchyma of mice exposed to CS and npBL, revealing that although alveolar spaces are similar to the control group, considerable leukocyte infiltration (arrowheads) is evident. In addition, septal exudate could be observed, confirming acute inflammation. Arrows indicate early stages of alveolar septa disruption, indicative of the onset of structural damage associated with CS exposure. Importantly, CS-exposed mice treated with the highest dose of npBX (CS + npBX 18%) showed similar histological characteristics as the control group, with preserved alveolar spaces and basal leukocyte levels and without any visible signs of inflammation (Figure 4C).

#### 3.3.2. npBX Prevents the Increase in Leukocyte Numbers and TNF-α Levels in BALF from Mice Exposed to CS

To investigate if npBX treatment was able to prevent the characteristic increase of leukocytes in BALF of mice with ALI, total and differential cell counts were performed (Figure 5A, Table 2). The CS + npBL group showed an increase in alveolar leukocytes when compared to the AA + npBL group. This was significantly reduced in a dose-dependent manner with npBX treatment, confirming the histological data that indicated inhibitory effects of npBX on CS-induced immune cell migration to the lung parenchyma. Differential cell counts indicated an increase in macrophages in BALF in CS + npBL, which could be prevented with the administration of the highest doses of npBX (12% and 18%). No significant differences could be detected in BALF neutrophil numbers between the AA + npBL and CS-exposed groups.

TNF-α is well-recognized as an important cytokine and chemokine driving the inflammatory process and promoting inflammatory cell migration. Figure 5B show that TNF-α BALF levels were increased in CS + npBL compared to AA + npBL animals. Treatment with npBX dose-dependently prevented increases in TNF-α, with a significant effect for 12% and a complete normalization to basal levels for 18% npBX.

#### 3.3.3. npBX Prevents Redox Imbalance Induced by CS Exposure in Mouse Lungs

Next, the redox homeostasis in mouse lungs after CS exposure for 5 consecutive days was evaluated. Figure 6A,B show total ROS levels and an estimate of NO (by nitrite (NO_2_)) in BALF, respectively. Both readouts present a similar pattern: CS + npBL showed an increase in these redox markers, which was prevented with npBX treatment in a dose-dependent manner. MDA is a marker directly related to oxidative stress [35]. MDA levels were significantly increased in lung homogenates from CS + npBL as compared to AA + npBL; as for redox markers in BALF, MDA elevations were dose-dependently prevented by treatment with npBX.

PNK is a marker of nitrosative damage and is formed as a result of peroxynitrite (ONOO−) nucleophilic attacks on protein tyrosine residues. Therefore, PNK protein expression was evaluated by Western blotting. Lung tissue homogenates from CS + npBL animals showed a modest increase in PNK expression compared to controls. In animals treated with npBX, PNK expression was maintained at basal levels (Figure 6D–E).

Evaluation of the endogenous antioxidant system was performed through SOD and CAT enzyme activity assays using lung tissue homogenates (Figure 7A,B). These demonstrated an increase in SOD activity and a reduction in CAT in the CS + npBL compared to the control group. This antioxidant disbalance was completely abolished with the highest dose of npBX (CS + npBX 18%). MPO activity was assessed in BALF as an inflammatory and redox marker once it can generate reactive chlorine species [45]. CS + npBL animals showed an increase in MPO activity compared to controls, which was effectively prevented when mice were treated with the highest dose of npBX (18%) (Figure 7C).

#### 3.3.4. npBX Antioxidant Activity is Independent of Nrf2

Nrf2 is a master regulator of the cellular antioxidant response. Considering ALI is characterized by a redox imbalance and bixin has previously been described as an Nrf2 inducer [35], the next step was to investigate whether npBX activates the Nrf2 pathway in this model. Immunohistochemistry revealed that alveolar leukocytes expressed Nrf2 in different cellular compartments (Figure 8). The percentage of areas expressing Nrf2 was increased in the CS + npBL group (Figure 8B), suggesting the occurrence of Nrf2 translocation as a response mechanism to counteract the redox imbalance caused by CS exposure. Interestingly, the percentage of positive areas stained for Nrf2 when animals were exposed to both CS and npBX decreased with increasing concentrations of npBX treatment.

#### 3.3.5. Bixin Nanoparticles Protected Lungs from MMP-9 Activity

Matrix metalloproteinase-9 (MMP-9) is a zinc-dependent enzyme with collagenase activity, and it is importantly involved in the establishment of various lung diseases [46]. The MMP-9 activity was evaluated in lung tissue homogenates by gelatin-gel zymography (Figure 9). Data showed that CS exposure induced MMP-9 activity, which could be completely prevented by treatment with npBX 18%.

## 4. Discussion

Oxidative stress plays an important role in the pathogenesis of ALI by affecting the lung parenchymal environment, promoting direct injury and through cellular as well as molecular mechanisms that drive lung inflammation and disrupt redox homeostasis [13]. The characteristic redox imbalance of ALI includes an increase in oxidative substances accompanied by a depletion of endogenous antioxidants. To evaluate the potential therapeutic effects of bixin on CS-induced inflammation and oxidative stress, mice were exposed to CS for five consecutive days and treated with three different doses of npBX (6, 12% and 18%). These findings suggest the application of npBX may be of therapeutic and pharmaceutical interest as treatment prevented the CS-induced redox imbalance and inflammation.

The npBXs showed negative ZP values, which potentiates their interaction with glycoproteins on the cell surface, thus, increasing their activity and endocytosis by the leukocyte cell system [47]. Another important piece of information refers to the lower ZP value; as mentioned before, PCL is an anionic polymer, conferring a negative charge for the NP formulations, and the obtained value difficult the aggregation of NPs due to electrostatic repulsion. NP size is a physical property that importantly affects cellular uptake, bloodstream clearance and route of administration [48]. Polymeric NPs with an MHD < 100 nm are usually more stable than larger ones, considering the latter are quickly opsonized by the immune system and eliminated by the reticuloendothelial system [49]. With regard to the route of administration, particles < 100 nm are capable of passing through epithelial barriers by diffusion, favoring oral bioavailability for enteric absorption [50].

A spherical morphology improves the hydrodynamic characteristics of NPs, facilitating transportation through the bloodstream. The combination of a spherical morphology and small size increases the cellular uptake of NPs by mononuclear phagocytes such as macrophages [51]. Considering our model of ALI, which is characterized by an imbalance in lung redox biology mainly driven by excessive macrophage activation [10], the administration of NPs with physical characteristics that promote interactions with mononuclear immune cells likely favors their therapeutic effectiveness.

The exothermic events depicted in the DSC curves suggest bixin stabilizes the NP formulation. In addition, the melting of bixin could be observed as an endothermic event at 170 °C, which was absent in npBX, indicating that the interaction with NPs could stabilize bixin in its crystalline structure even at high temperatures.

CS-induced ALI is characterized by an influx of inflammatory cells, such as macrophages and neutrophils, into the lungs. These inflammatory leukocytes are activated by toxic compounds present in CS and increase the production of ROS [12,16,52,53,54]. This is the first study to evaluate bixin as a pharmaceutical formulation in NPs for the treatment of ALI induced by CS exposure in mice. The presented model adequately represents the ALI condition as indicated by an increase in alveolar leukocytes and ROS levels in CS + npBL mice. Interestingly, lymphocyte count was lower in the CS + npBL group, possibly due to the priority of recruitment of macrophages at this stage of inflammation. Interestingly, in the differential leukocyte count, neutrophil levels did not show a significant difference between groups, in special when comparing AA + npBL and CS + npBL. Although MPO activity showed an increase in the CS + npBL group compared to AA + npBL, this could mean that these neutrophils are being stimulated in some way by the cigarette smoke (CS) and becoming more activated, producing more MPO, but not being recruited with the same intensity, the underlying mechanism needs further investigations. This data is corroborated by another study which showed similar lymphocyte numbers in a model of CS exposure with wild-type mice [55]. Importantly, treatment with 12% and 18% npBX completely prevented the CS-induced increase in ROS levels and leukocyte numbers in BALF. This point out that bixin has strong antioxidant and anti-inflammatory effects in this ALI model. The observation that the increase in ROS production is parallel to the number of alveolar leukocytes is not surprising, considering activated immune cells promote oxidative burst beyond the production and release of inflammatory mediators [8].

Lee et al. (2016) reported that treatment with an antioxidant molecule could prevent an increase in BAL leukocyte levels as well as the production of ROS in mice exposed to CS, thereby preventing inflammation, decreasing ROS/NF-κB pathway activity, and promoting antichemotactic activity (decrease in chemotaxis factors IL-6 and TNF-α) [56]. In addition, Valdivieso et al. (2018) showed that treatment with an antioxidant decreased IL-6 and IL-8 production by Calu-3 cells exposed to CS extract (CSE) in vitro [57]. In line with these observations, the present data suggest bixin, a known antioxidant agent, also modulates inflammatory cytokine production as indicated by the dose-dependent reduction (and full normalization to control levels with 18% npBX) of CS-induced TNF-α levels in BALF.

ONOO− itself is an unstable oxidative agent generated from the reaction between nitric oxide and superoxide anion [58]. Its presence in tissue leads to oxidative damage to proteins, marked by nitration reaction on tyrosine residues. The increase in NO levels (measured by nitrite) in the CS + npBL group was absent in animals treated with npBX. Both ROS and NO results collectively support the increase in nitrotyrosine (PNK) expression detected in lung tissue from the CS + npBL group. NpBX completely avoided this oxidative damage.

Oxidative stress was assessed by measuring the levels of ROS, the antioxidant enzymatic activities of SOD/CAT, and the oxidative damage marker MDA, from lipid peroxidation, beyond the peroxynitrite mentioned above. Treatment of animals with npBX during the 5-day CS exposure protocol preserved the ROS levels as well as SOD and CAT enzymatic activities, effects that were most pronounced when 18% (*w*/*v*) npBX was given. Additionally, our results also point to npBX as a complex capable of preventing lipid peroxidation that occurs in the presence of ROS, as indicated by reduced MDA levels in treated animals. SOD is the primary enzymatic defense against anion superoxide by neutralizing O_2_^•−^ to H_2_O_2_, which is neutralized to water by CAT [15]. CS exposure promoted an increase in SOD and a decrease in CAT activities in the CS + npBL group, suggesting anion superoxide was neutralized to hydrogen peroxide that could not be effectively metabolized by CAT, accumulating in lung tissue and contributing to oxidative damage. Treatment with npBX was able to neutralize the oxidative molecules generated in response to CS exposure, considering antioxidant activities of SOD/CAT were fully preserved in the CS + npBX 18% group. Altogether, these findings strongly suggest bixin, when administered loaded on NPs, is an effective antioxidant.

In the present CS-exposure model, animals demonstrated a redox imbalance due to a disruption in the endogenous antioxidant enzymatic system. In the CS + npBL group, there was a simultaneous increase in the O_2_^•−^ and NO levels, leading to the formation of ONOO- that is responsible for the nitration of tyrosine protein residues. In the presence of bixin, this reaction does not occur because bixin is able to neutralize the O_2_^•−^ preventing nitrosative damage.

Interestingly, the percentage of positive cells for nuclear Nrf2 was decreased in animals treated with npBX as compared to CS + npBL mice, despite all the antioxidant effects observed. This result suggests that npBX acts upstream of Nrf2 activation, supporting the scavenger role for bixin.

MMP-9 is a zinc-dependent enzyme with collagenase activity importantly involved in the establishment of various lung diseases [46]. The activity of MMP-9 by gelatin gel zymography assay showed that CS induced an increase in MMP-9 activity, which could be prevented by concomitant npBX 18% treatment. Under inflammatory conditions, resident cells and leukocytes can produce MMP-9 [59], and its enzymatic activity can be upregulated under conditions of oxidative stress (increased ROS levels) [60]. In addition, MMP9 plays an important role in the production of proinflammatory cytokines, such as TNF-α [61]. Thus, npBX was able to prevent the CS-induced increase in MMP-9 activity, thereby likely reducing the production of TNF-α and migration of macrophages, ultimately resulting in decreased ROS production. Because MMP-9 is responsible for the degradation of the extracellular matrix, modulating its activity is fundamental to preventing the progression of chronic pulmonary diseases induced by CS. In this regard, an early onset of alveolar structural destruction after 5-day CS exposure was observed, indicative of the development of emphysema, which was strongly prevented by npBX treatment.

Overall, these data suggest the use of bixin-loaded NPs (particularly in the higher dose range (18% npBX)) can be considered an attractive therapeutic strategy for the prevention of lung diseases associated with CS exposure, and future studies will be aimed at validating these findings using human tissue and evaluating the effects of npBX in models of chronic obstructive pulmonary disease. Access to other routes of administration, such as intranasal, and comparing the results with the NP-free bixin is also required.

## 5. Conclusions

Simultaneous treatment with npBX and exposure to CS effectively prevented CS-induced damages in an apparent Nrf2-independent fashion. The beneficial effects of npBX treatment can likely be attributed to the ability of bixin to scavenge and neutralize oxidative substances, blocking the harmful successive events triggered by CS.

## Figures and Tables

**Figure 1 antioxidants-11-01293-f001:**
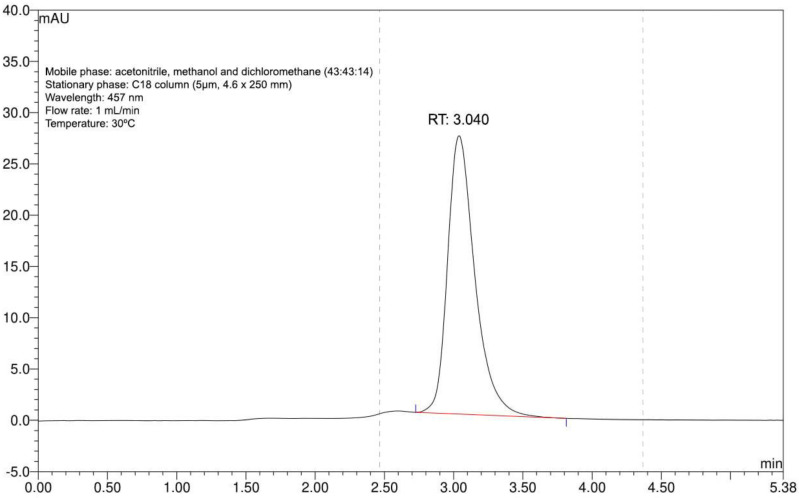
Annatto seed extract chromatogram. Identity confirmation of bixin with compliance retention time (RT) at 457 nm and standardized chromatographic parameters.

**Figure 2 antioxidants-11-01293-f002:**
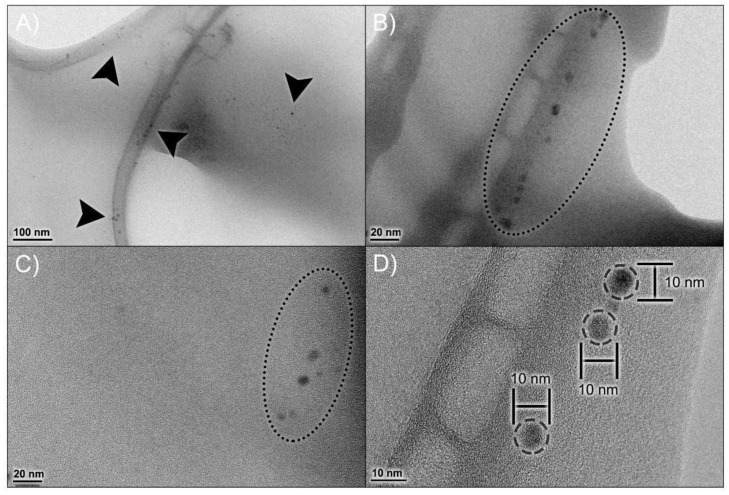
Morphological profile of npBX. Transmission Electron Microscopy (TEM) micrographs of npBX at various magnifications (**A**–**C**), showing the size distribution of npBX and their consistent diameter (**D**). Images were captured with a JEOL 1200 EX (Akishima, Tokyo, Japan) at 80 kV without staining.

**Figure 3 antioxidants-11-01293-f003:**
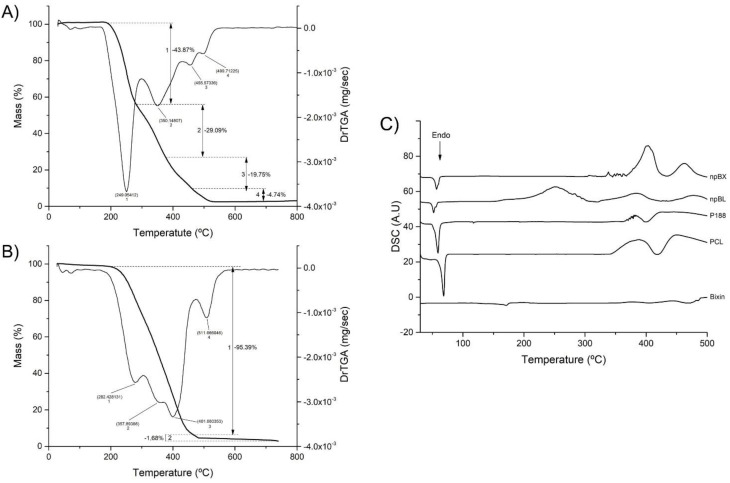
Thermal stability of nanoparticles. (**A**) TGA/DrTGA from 0 to 800 °C of npBL. (**B**) TGA/DrTGA from 0 to 800 °C of npBX. (**C**) DSC curves of bixin, PCL, P 188, npBL and npBX.

**Figure 4 antioxidants-11-01293-f004:**
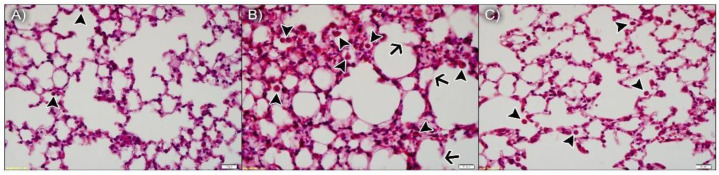
Bixin-loaded nanoparticles protect against CS-induced leukocyte infiltration and onset of structural damage in the lung parenchyma. Histological slides stained with H&E from AA + npBL (**A**), CS + npBL (**B**) and CS + npBX 18% (**C**) groups. Arrowheads indicate leukocyte infiltration. Arrows represent early stages of alveolar septa disruption. Scale bar: 20 µm. Original magnification: 40×.

**Figure 5 antioxidants-11-01293-f005:**
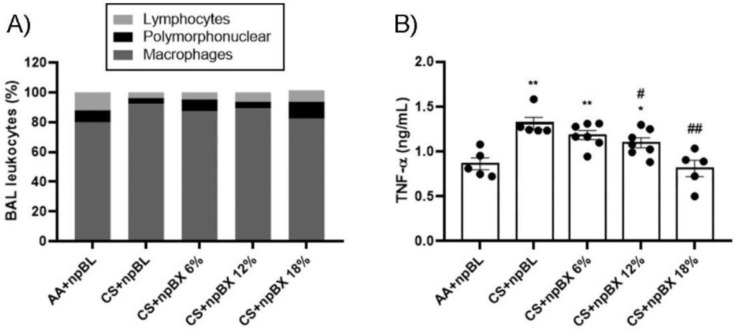
Anti-inflammatory effects of bixin-loaded nanoparticles. (**A**) Total and differential cell counts in BALF. (**B**) TNF-α levels in BALF samples from 8 animals per group as determined by ELISA (AA + npBL *n* = 5; CS + npBL *n* = 5; CS + npBX 6% *n* = 7; CS + npBX 12% *n* = 7; and CS + npBX 18% *n* = 5). * *p* < 0.05 and ** *p* < 0.01 compared to AA + npBL; # *p* < 0.05 and ## *p* < 0.01 compared to CS + npBL.

**Figure 6 antioxidants-11-01293-f006:**
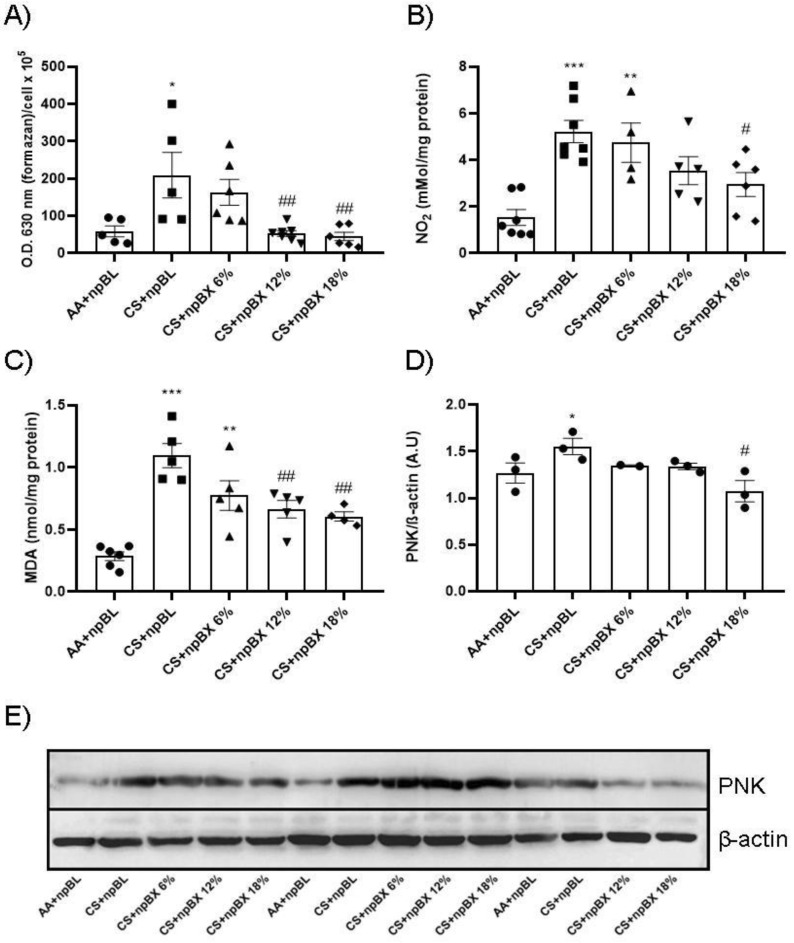
Effects of bixin-loaded nanoparticles on markers of the redox homeostasis in the lung. (**A**) Total ROS in BALF determined by NBT method. (**B**) Nitrite (NO_2_) levels in BALF as a nitric oxide (NO) marker measured by Griess assay. (**C**) Lung homogenate levels of MDA as a lipid oxidative damage marker. (**D**) Densitometry of Western blotting (*n* = 3) results for PNK expression (in lung homogenates). (**E**) Representative image of Western blotting for PNK and β-actin. * *p* < 0.05, ** *p* < 0.01, *** *p* < 0.001 when compared to AA + npBL; # *p* < 0.05 and ## *p* < 0.01 when compared to CS + npBL.

**Figure 7 antioxidants-11-01293-f007:**
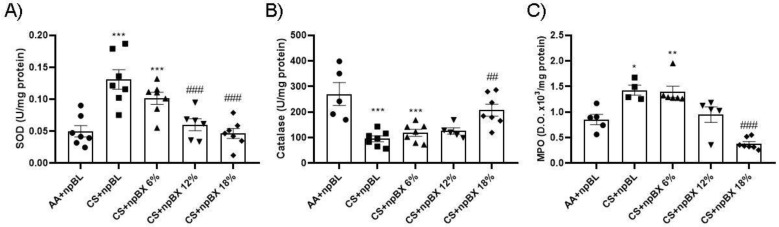
Treatment with bixin-loaded nanoparticles prevents CS-induced disturbances in the endogenous antioxidant response. Superoxide dismutase (**SOD**, **A**) and catalase (CAT, **B**) activities were assessed in lung tissue homogenates. MPO activity (**C**) was determined in BALF. * *p* < 0.05, ** *p* < 0.01, *** *p* < 0.001 compared to AA + npBL; ## *p* < 0.01 and ### *p* < 0.001 compared to the CS + npBL group.

**Figure 8 antioxidants-11-01293-f008:**
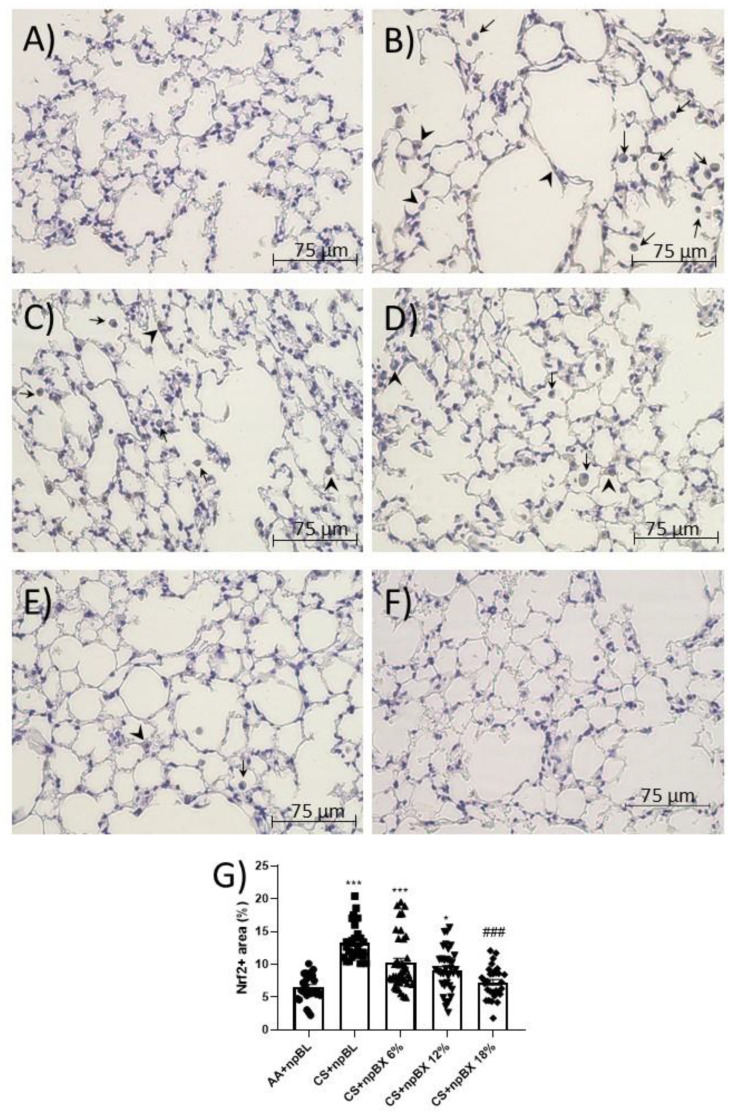
Immunohistochemistry for Nrf2. Lung sections stained for Nrf2 of (**A**) AA + npBL, (**B**) CS + npBL, (**C**) CS + npBX 6%, (**D**) CS + npBX 12%, (**E**) CS + npBX 18% and (**F**) negative control (lacking the primary antibody) mice. (**G**) Quantification of positive cells. Arrows and arrow heads indicate Nrf2^+^ cells into the alveolar space and in alveolar sept, respectively. Scale bar: 75 µm. Original magnification: 40x. * *p* < 0.05 and *** *p* < 0.001 compared to AA + npBL; ### *p* < 0.001 compared to the CS + npBL group.

**Figure 9 antioxidants-11-01293-f009:**
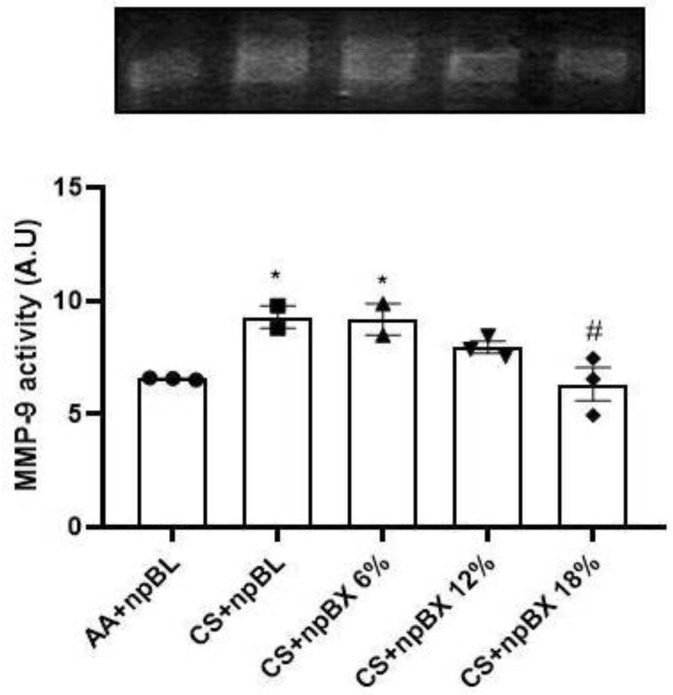
Bixin-loaded nanoparticles inhibit CS-induced MMP-9 activity in lung homogenates. MMP-9 enzyme activity determined by SDS-PAGE gel zymography with 1.5 mg/mL gelatin. * *p* < 0.05 compared to AA + npBL; # *p* < 0.05 compared to CS + npBL.

**Table 1 antioxidants-11-01293-t001:** Physical characterization of npBX.

	DLS			TEM
	Medium Hydrodynamic Diameter (nm)	Polydisperssivity Index	Zeta Potential (mV)	Mean Diameter (nm)
npBX	23.57 ± 0.17	0.23 ± 0.01	−31.73 ± 10.10	6.51 ± 0.34

**Table 2 antioxidants-11-01293-t002:** Total and differential leukocyte cell counts in BALF samples from 8 animals per group. * *p* < 0.05 and *** *p* < 0.001 compared to AA + npBL; ## *p* < 0.01 and ### *p* < 0.001 compared to CS + npBL; $ *p* < 0.05, $$ *p* < 0.01, $$$ *p* < 0.001 compared to CS + npBX 6%.

Group	Cell × 10^5^ Mean ± SEM (%)			
	Total	Macrophages	Neutrophils	Lymphocyte
AA + npBL	3.08 ± 0.47 (100)	2.51 ± 0.35 (79.78)	0.23 ± 0.06 (8.11)	0.34 ± 0.06 (12.11)
CS + npBL	6.24 ± 0.88 (100) ***	5.77 ± 0.80 (92.15) ***	0.26 ± 0.05 (4.14)	0.21 ± 0.03 (3.71) *
CS + npBX 6%	4.94 ± 0.9 (100) *	4.36 ± 0.76 (87.5) ***	0.37 ± 0.09 (7.83)	0.21 ± 0.05 (4.67) *
CS + npBX 12%	3.84 ± 0.53 (100) ^##^	3.44 ± 0.45 (89.19) ^### $$^	0.17 ± 0.04 (4.56) ^$^	0.23 ± 0.04 (6.27)
CS + npBX 18%	3.26 ± 0.7 (100) ^### $^	2.64 ± 0.48 (82.19) ^### $$$^	0.36 ± 0.15 (11.66)	0.26 ± 0.07 (7.6)

## Data Availability

The data is contained within this article.

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
