# Peer review of "Treatment with Bixin-Loaded Polymeric Nanoparticles Prevents Cigarette Smoke-Induced Acute Lung Inflammation and Oxidative Stress in Mice"

_antioxidants, 2022, doi:10.3390/antiox11071293_

Round 1

Reviewer 1 Report

The authors report prevention of cigarette smoke-induced acute lung inflammation by co-treatment with bixin-loaded polymeric nanoparticles in mice. Treatment with bixin-loaded nanoparticles completely prevented ROS and nitrite production parallel with reducing leukocyte number and TNF levels in lavage fluid as well as the expression of Nrf2. While these findings may imply therapeutic potential for bixin, there are several issues that require further attention.

1. It is uncertain whether the beneficial actions of bixin can be attributed to antioxidant or anti-inflammatory effects or both. The authors speculate about inhibition of ONOO- formation without documenting enhanced NO formation in their model.

2. The inhibitory action of bixin-loaded nanoparticles on Nrf2 expression is intriguing. What is the underlying mechanism? Along this line, the Abstract may be misleading for it refers to inhibition of Nrf2 expression as well as to an effect “independent of an increase of Nrf2 exprssion”. These should be clarified.   

3.  Some of the experimental protocols are difficult to follow. It is unclear when treatment with bixin-loaded nanoparticles was initiated (i.e. before exposure of animals to cigarette smoke?). A clinically more relevant approach will be to induce ALI first and then initiate treatment to assess whether lung damage can be reversed.

4. What was the rationale for conserving BAL fluid on dry ice before centrifugation? Were all lungs lavaged first before submitting the left lungs to tissue processing? How could then the authors account for removal of inflammatory cells from the alveolar space?

5. Annatto seed extraction and characterization appear to have already been published by the authors (ref #38). What is the rationale for repeating this information? Furthermore, Figure 1 lack legends and Table 1 is superfluous, for the same data are also presented in the text.

6. The dose-dependency of bixin-loaded nanoparticles on neutrophil counts is peculiar, for statistically significant decreased were detected with only 12% npBX. There seems to be increases in neutrophil count with npBX at 6 or 18% over control values. How could these be reconciled with an anti-inflammatory action?

7. The legend to Figure 5 refers to 8 animals per group, whereas 5-7 individual data points are shown on the figure. Which number is correct?

8. Were lung homogenates assessed for PNK after lavage?

9. What was the rationale of analyzing MPO in BAL fluid only, and not in the lung tissue (i.e. as an index of tissue-infiltrating neutrophils)?

10. The immunohistochemistry images are difficult to assess. It is unclear how was positive cells defined. An essential control (staining with secondary Ab only) is missing.

11. Whole lung lysates do not allow assessment of cellular source(s) of MMP9. What evidence indicates that macrophages were the sole source of MMP9? Bixin-loaded nanoparticles reduced leukocyte numbers in a dose-dependent manner, whereas statistically significant decreased MMP9 activity was detected with npBX 18% only. How could these observations be reconciled?

Author Response

Dear Reviewer, the authors would thankful for all of your considerations, once we believe a more detailed explanation will contribute to improving the clarity of our study, especially for those that do not work with pulmonary-derived samples. A point-by-point response can be found in attached PDF file.

Reviewer 2 Report

Antioxidants

Manuscript ID: antioxidants-1781638

Title: Treatment with bixin-loaded polymeric nanoparticles prevents cigarette smoke-induced acute lung inflammation and oxidative stress in mice

The topic of the article is interesting, important, and fitting for the journal Antioxidants. A carotenoid from a native plant, named bixin, was shown to have beneficial effects against oxidative stress and inflammation induced in mice exposed to cigarette smoke. Due to its hydrophobic properties, the compound was encapsulated in polymeric nanoparticles. Overall, the article reads well, I only have a few comments, questions that would improve it.

1.       What does this 18% refer in blank nanoparticle (npBL)?

2.       2.7. Experimental procedure. Please indicate the volume that was fed to the mice through the gavage.

3.       It is puzzling that the neutrophil count did not differ between groups AA and CS (Table 2). Could you please comment on this in the discussion.

Author Response

Dear Reviewer. Thanks for your considerations. Bellow, a poin-by-poin answer can be find adressing all of your comments and highlighting all modifications in the main text.

Point 1: What does this 18% refer in blank nanoparticle (npBL)?

Response 1. The 18% npBL refers to the concentration in w/v of the administrated suspension. In our model, the blank nanoparticles (polymers-based only) are the vehicle for bixin. To prove that the beneficial finds refer only to the effects of the active pharmaceutical ingredients (API – bixin) and not to its excipients (polymers) we decided to use the highest concentration tested for npBX (bixin-loaded polymeric nanoparticles) for npBL too, thus, excluding any detrimental effect from the vehicle.

Point 2: 2.7. Experimental procedure. Please indicate the volume that was fed to the mice through the gavage.

Response 2. We are so sorry for any lack in our methodology, the following sentence was added to the manuscript:

“The treatment was administered daily by gavage in a total volume of 100 µL each, in order to evaluate if the colloidal suspension of npBX would keep bixin antioxidant properties”.

Point 3: It is puzzling that the neutrophil count did not differ between groups AA and CS (Table 2). Could you please comment on this in the discussion.

Response 3. Our data showed an increase in MPO (myeloperoxidase) activity without a significant neutrophil infiltration, this could mean that these neutrophils are being stimulated in some way by the cigarette smoke (CS) and getting more activated, producing more MPO, but not being recruited with the same intensity. The following sentence will be added to the discussion topic of the manuscript:

“Interestingly, in the differential leukocyte count, neutrophil levels did not show a significant difference between groups, in special when comparing AA+npBL and CS+npBL. Although MPO activity showed increased in the CS+npBL group compared to AA+npBL, this could mean that these neutrophils are being stimulated in some way by the cigarette smoke (CS) and getting more activated, producing more MPO, but not being recruited with the same intensity, the underlying mechanism needs further investigations.”

Tring to explore this fenomenum, the intercellular adhesion molecule 1 (ICAM-1) was investigated by western blotting, showing a dose-dependent reduction. But this data is inconclusive and preliminary, needing to be reproduced with more samples. ICAM-1 is a protein mainly related to the leukocyte migration process, regulating the rolling and adhesive process interactions with endothelial cells, and guiding leukocytes to cross the endothelial layer. This reduction in ICAM-1 expression could be one of the anti-inflammatory mechanisms performed by bixin, explaining the low neutrophil count in group CS+npBX 18%, since they are not being recruited to the alveoli.
